# Genome-Wide Identification, Evolutionary Analysis, and Expression Patterns of Cathepsin Superfamily in Black Rockfish (*Sebastes schlegelii*) following *Aeromonas salmonicida* Infection

**DOI:** 10.3390/md20080504

**Published:** 2022-08-03

**Authors:** Yuqing Li, Xingchun Li, Pei Zhang, Defeng Chen, Xinran Tao, Min Cao, Chao Li, Qiang Fu

**Affiliations:** School of Marine Science and Engineering, Qingdao Agricultural University, Qingdao 266109, China; liyuqing@stu.qau.edu.cn (Y.L.); lixingchun@stu.qau.edu.cn (X.L.); zhangpei@stu.qau.edu.cn (P.Z.); chendefeng@stu.qau.edu.cn (D.C.); taoxinran@stu.qau.edu.cn (X.T.); caomin@qau.edu.cn (M.C.); chaoli@qau.edu.cn (C.L.)

**Keywords:** cathepsin, black rockfish, evolution rates, innate immunity, expression profile

## Abstract

Cathepsins are lysosomal cysteine proteases belonging to the papain family and play crucial roles in intracellular protein degradation/turnover, hormone maturation, antigen processing, and immune responses. In the present study, 18 cathepsins were systematically identified from the fish *S. schlegelii* genome. Phylogenetic analysis indicated that cathepsin superfamilies are categorized into eleven major clusters. Synteny and genome organization analysis revealed that whole-genome duplication led to the expansion of *S. schlegelii* cathepsins. Evolutionary rate analyses indicated that the lowest Ka/Ks ratios were observed in CTSBa (0.13) and CTSBb (0.14), and the highest Ka/Ks ratios were observed in CTSZa (1.97) and CTSZb (1.75). In addition, cathepsins were ubiquitously expressed in all examined tissues, with high expression levels observed in the gill, intestine, head kidney, and spleen. Additionally, most cathepsins were differentially expressed in the head kidney, gill, spleen, and liver following *Aeromonas salmonicida* infection, and their expression signatures showed tissue-specific and time-dependent patterns. Finally, protein–protein interaction network (PPI) analyses revealed that cathepsins are closely related to a few immune-related genes, such as interleukins, chemokines, and TLR genes. These results are expected to be valuable for comparative immunological studies and provide insights for further functional characterization of cathepsins in fish species.

## 1. Introduction

Lysosomal proteases exert vital roles in several highly regulated life processes in both prokaryotes and eukaryotes, including cell division, fertilization, vitellogenesis, differentiation, tissue remodeling, antigen processing, and apoptosis [1,2,3]. The lysosome is a membrane-bound cytoplasmic organelle that serves as a major degradative compartment in eukaryotic cells [4]. Both endogenous and exogenous macromolecules can be delivered to lysosomes through biosynthetic and endocytic pathways, respectively. In addition, lysosomes can degrade proteins transported from the cytosol [3]. Therefore, the lysosome plays an important role in normal metabolism to maintain cellular homeostasis. Four catalytic groups of proteolytic enzymes (serine, cysteine, aspartic, and metalloproteases) have been recognized on the basis of the reaction nucleophile in the catalytic site of the enzyme [5]. The degradative function of the lysosome is carried out by more than 50 acid-dependent hydrolases (e.g., proteases, lipases, and glycosidases) contained within its lumen, which has an acidic pH in the range of 4.6–5.0 [6].

Cysteine proteases of the papain superfamily, which are regularly referred to as lysosomal cathepsins, comprise a large number of enzymes [7]. On the one hand, cathepsins can be divided into three subgroups, namely, cysteine proteases (cathepsins B, C, F, H, K, L, O, S, T, U, V, W, and X), serine proteases (cathepsins A and G), and aspartic proteases (cathepsins D and E), according to the amino acid residues in their active sites [8,9]. Among them, cysteine cathepsins can be further classified into two subgroups based on sequence similarity, length, and structure of the pro-region: enzymes with the highly conserved interspersed ERFNIN (E-X3-R-X3-F-X2-N-X3-I-X-N) motif within the propeptide region, such as cathepsin L-like proteases (cathepsins F, H, K, L, S, V, and W), and those with shorter propeptides lacking this motif, including cathepsin B-like cysteine proteases (cathepsins B, C, O, and X) [10,11]. On the other hand, according to the tissue distribution, cathepsins B, C, F, H, L, O, and Z compose the ubiquitously expressed group, and cathepsins K, W, and X are cell-specifically or tissue-specifically expressed [12]. Cysteine cathepsins can form a V-shaped substrate-binding cleft through a papain-like fold, and variations in shape and residues make cysteine cathepsin members present differences in substrate preference and catalytic properties [8,13].

In mammals, the immune function of the cathepsin family has been extensively reported. Cathepsin A was reported to exert roles in chaperone-mediated autophagy and the cellular oxidative stress response [14,15]. Cathepsin B, a unique member of the papain superfamily, plays important roles in pathological and physiological processes, such as apoptosis [16], cancer, the TLR9 signaling pathway [17], and TNF-α posttranslational processing in macrophages [18]. Cathepsin C is a central coordinator for the activation of various serine proteinases in inflammatory and immune cells [19]. Contrary to other proteases, cathepsin D seems to be more of a mitogen than a protease, allowing cancer cells to cross the basement membrane and invade connective tissue or enter the bloodstream [20]. Cathepsin H plays important roles in the intracellular degradation of proteins and antigen processing [21] and is also required for disease treatment and lung development [22]. Cathepsin K stands out for several reasons, such as its central role in bone resorption, bone erosion, and immune regulation [23]. Cathepsin L has been revealed to play vital roles in controlling normal mucosal epithelial homeostasis and supporting the host immune defense against infection in mammal mucosal surfaces [24]. Cathepsin O is widely expressed in human tissues and has been suggested to play a role in normal cellular protein degradation and turnover [25]. High expression levels of CTSS have been reported in many immune cells, especially during inflammatory responses, including B-lymphocytes, lysosomes of macrophages, antigen-presenting cells, dendritic cells, and epithelial cells [26,27,28]. Cathepsin Z plays an important housekeeping role as a lysosomal digestive enzyme, similar to cathepsins B, L, H, and O in Human [29]. However, the functional research on cathepsins in fish species is very limited, since most of the previous studies focused on their identification and expression signatures. For example, the cloning and expression analysis of cathepsin S was conducted during vitellogenesis and oocyte maturation in killifish (*Fundulus heteroclitus*) [30]. Phylogenetic analyses and characterization of cathepsins such as CTSL and CTSH were performed in the channel catfish (*Ictalurus punctatus*) [31] and cichlid (*Paralabidochromis chilotes*) [3]. The expression analysis and enzymatic characterization of cathepsin S were reported in the olive flounder (*Paralichthys olivaceus*) [32].

Black rockfish is an economically important fish species widely cultured in South Korea, Japan, and China. Recently, *S. schlegelii* has been continuously threatened by bacterial infections, such as *Vibrio anguillarum*, *Edwardsiella tarda*, and *Aeromonas salmonicida*, causing great economic losses to the farming industry of *S. schlegelii* [33,34]. In particular, *A. salmonicida* is composed of a collection of Gram-negative bacteria, widespread in water environments [35]. However, the immune mechanism of black rockfish in response to pathogen infection is very limited. As cathepsins play vital roles in immune responses to bacterial infections, clarification of the molecular mechanism for cathepsins in response to *A. salmonicida* infection is expected to promote the development of effective strategies in bacterial disease management for *S. schlegelii*. Previous studies reported three members of the cathepsin family (CTSB, CTSK, and CTSSb) in *S. schlegelii* [23,36,37]. In this study, we first systematically identified all possible members of the cathepsin superfamily in *S. schlegelii*. In addition, we investigated their expression in vivo in healthy *S. schlegelii* tissues and their responses in the head kidney, liver, gill, and spleen tissues following infection with *A. salmonicida*. These results are expected to provide fundamental support for a better understanding of the immune functions of teleost cathepsins in response to bacterial infection.

## 2. Results

### 2.1. Identification of S. schlegelii Cathepsins

In the present study, a total of eighteen cathepsins were identified in the *S. schlegelii* genome, named CTSAa, CTSAb, CTSBa, CTSBb, CTSC, CTSDa, CTSDb, CTSF, CTSHa, CTSHb, CTSK, CTSLa, CTSLb, CTSO, CTSSa, CTSSb, CTSZa, and CTSZb (Table 1). Their characteristics, including names, coding sequence lengths, isoelectric points (PIs), molecular weights (MWs), gene orientations, chromosomal locations, instability indices, aliphatic indices, and formulas, are summarized in Table 1. In particular, the coding sequence lengths, PIs, MWs, instability indices, and aliphatic indices of the eighteen cathepsins varied from 215 aa (CTSHb) to 640 aa (CTSSa), from 5.02 (CTSLb) to 8.41 (CTSDb), from 24.66 (CTSHb) to 71.66 (CTSSa), from 29.14 (CTSLa) to 42.11 (CTSHb), and from 60.00 (CTSHa) to 94.07 (CTSDb), respectively. In addition, their tertiary structures were constructed using the Phyre2 server and colored by rainbow from the N- to C-terminus (Figure 1).

### 2.2. Gene Structure Analysis of S. schlegelii Cathepsins

The structural characteristics of cathepsins were investigated, including the numbers of exons and motif patterns (Figure 2). On the one hand, the exon numbers of most cathepsins are relatively stable, varying between 8 and 12, except for CTSZb (5 exons), CTSZa (7 exons), and CTSAa (13 exons) (Figure 2B). On the other hand, the motif patterns of most cathepsins are relatively conserved. Specifically, CTSBa, CTSBb, CTSC, CTSF, CTSHa, CTSHb, CTSK, CTSLa, CTSLb, CTSO, CTSSa, CTSSb, CTSZa, and CTSZb all possess the functional domain Pept_C1 domain, and nine of them harbor the cathepsin propeptide inhibitor I29 domain. In addition, both CTSAa and CTSAb possess the Peptidase_S10 domain, and both CTSDa and CTSDb possess the Taxi_N domain (Figure 2C).

### 2.3. Phylogenetic and Synteny Analysis of S. schlegelii Cathepsins

To validate the identification of the cathepsins, as well as explore the evolutionary relationship among different organisms, phylogenetic analysis was performed (Figure 3). Overall, the phylogenetic relationship of cathepsins is relatively conserved, supporting the annotation of *S. schlegelii* cathepsins (Figure 3). Most of the cathepsins were clustered into clades with their respective counterparts of flounder, medaka (*Oryzias latipes*), fugu (*Takifugu rubripes*), and turbot (*Scophthalmus maximus*) and then clustered with other teleost and tetrapod species, with strong bootstrap support. The only exception was CTSLb, which was clustered with CTSS and CTSZ of zebrafish (*Danio rerio*) and then grouped into the clade of CTSLb of zebrafish. In addition, cathepsin genes were divided into eleven major subgroups: CTSA, CTSB, CTSC, CTSD, CTSF, CTSH, CTSL, CTSK, CTSO, CTSS, and CTSZ. Of the eighteen cathepsins, a number of them are highly related duplicates. For example, CTSA, CTSB, CTSD, CTSH, CTSL, CTSS, and CTSZ have two copies (Figure 3).

Though phylogenetic relationships provided strong support for the identification of most cathepsins, syntenic analyses were performed to provide additional evidence (Figure 4). On the one hand, the synteny analyses provided strong evidence for the identification of *S. schlegelii* CTSLb, whose phylogenetic relationship was obscure. The conserved flanking genes included parkin co-regulated gene (*pacrg*), quaking-a (*qkia*), (*ythdf2*), arfgap with sh3 domain, ankyrin repeat and PH 3 (*asap3*), connector enhancer of kinase suppressor of ras 1 (*cnksr1*), and grainyhead-like 3 (*grhl3*) among black rockfish, zebrafish, and catfish (Figure 4M). On the other hand, the eighteen *S. schlegelii* cathepsins identified in this study were all well clarified and annotated. For instance, the neighboring genes of CTSBa, including fanconi anemia complementation group M (*fancm*), set and mynd Domain Containing 2a (*smyd2a*), German-Chinese forum on computational catalysis 2 (*gcfc2*), transcription factor B2, mitochondrial (*tfb2m*), potassium channel tetramerization domain-containing 3 (*kctd3*), and estrogen-related receptor gamma (*esrrg*), were conserved among black rockfish, zebrafish, and channel catfish (Figure 4C). The neighboring genes of CTSF, including mitochondrial ribosomal protein L18 (*mrpl18*), transient receptor potential vanilloid 1 (*trpt1*), heterogeneous nuclear ribonucleoprotein H1 (*hnrnph1*), run and fyve domain-containing 1 (*rufy1*), spermatogenesis-associated gene 4 (*spata4*), platelet-derived growth factor C (*pdgfc*), and Translocase of outer mitochondrial membrane 5 (*tomm5*), were conserved among black rockfish, zebrafish, and catfish (Figure 4H). The adjacent gene (*rassf9*) downstream of CTSAb in zebrafish and channel catfish was also observed downstream of the CTSAb gene in black rockfish. Moreover, the synteny analysis supported the annotations of duplicated genes, including CTSA, CTSB, CTSD, CTSH, CTSL, CTSS, and CTSZ. In detail, CTSA, CTSB, CTSD, CTSH, CTSL, CTSS, and CTSZ were all duplicated by whole-genome duplication.

### 2.4. Genomic Clusters of S. schlegelii Cathepsins

A total of eighteen cathepsins were distributed on eleven chromosomes (chromosomes 1, 2, 5, 6, 8, 9, 14, 15, 16, 17, and 22). In detail, chromosome 2 contained the largest number of cathepsins, with four genes: CTSDa, CTSBa, CTSHb, and CTSAb. In addition, there were three genes located on chromosome 8, and chromosome 1 and chromosome 22 each had two genes. Meanwhile, other genes were distributed individually on seven different chromosomes, whereas none were located on chromosomes 3, 7, 10, 12,13, 17, 18, 19, 21, or 23 (Figure 5 and Figure 6).

### 2.5. Evolutionary Rate Analysis

To determine whether cathepsins evolved under different selection pressures among the selected teleost species, the values of Ka, Ks, and their ratio for each homolog from nine selected teleost species were compared. As shown in Figure 7, the values of Ka among these cathepsins ranged from 0.15 to 0.86, and the values of Ks varied from 0.16 to 1.41. In addition, cathepsins were grouped into two categories according to their ratio of Ka to Ks. The average values of Ka/Ks of CTSC, CTSDa, CTSF, CTSHa, CTSHb, CTSO, CTSSa, CTSSb, CTSZa, and CTSZb were greater than one, indicating that these genes were under positive selection. Among them, the highest Ka/Ks ratios were observed in CTSZa and CTSZb, with values of 1.97 and 1.75, respectively. The Ka/Ks ratios of other cathepsins were less than one, revealing that they underwent efficient purifying selection. The lowest Ka/Ks ratios were observed in CTSBa (0.13) and CTSBb (0.14).

### 2.6. Expression Profiles of Cathepsins after A. salmonicida Infection

The expression patterns of *S. schlegelii* cathepsins were detected in nine healthy tissues, including the blood, brain, gill, intestine, liver, spleen, head kidney, skin, and muscle. Among the tested tissues, the cathepsins were expressed at comparatively low levels in the blood, and then we used the expression level in the blood as the baseline for *S. schlegelii* cathepsins. Overall, *S. schlegelii* cathepsins were ubiquitously expressed in all of the examined tissues with distinct expression patterns (Figure 8). In detail, the highest expression levels of most *S. schlegelii* cathepsins were observed in the gill, head kidney, and spleen, while the lowest expression levels were found in the blood, muscle, and liver, with moderate expression levels observed in the brain, intestine, and skin (Figure 8).

To further understand the involvement of cathepsins in response to bacterial infection, *S. schlegelii* were experimentally challenged with *A. salmonicida*, and the expression patterns of *S. schlegelii* cathepsins were explored in the gill (Figure 9), head kidney (Figure 10), liver (Figure 11), and spleen (Figure 12) by qPCR. In general, all of the *S. schlegelii* cathepsins were significantly differentially expressed, and most of them were significantly up-regulated in those four tissues. In the gill, all of the *S. schlegelii* cathepsins showed a significant up-regulation trend at all time points, except for CTSSb at 6 h (Figure 9). Among them, CTSK was up-regulated the most, with an expression level change of 303.21-fold at 6 h. CTSHa was up-regulated by more than one-hundred-fold at all four time points. Similarly, all of the *S. schlegelii* cathepsins were significantly up-regulated in the head kidney at all time points, except for CTSBb and CTSSb (Figure 10). CTSBb was significantly down-regulated at 48 h and 72 h, and CTSSb was significantly down-regulated at 72 h. Besides three genes (CTSDb at 24 h, CTSHa except for at 72 h, and CTSLb at 24 h), all genes were differentially expressed with fold changes of less than fifty in the kidney. In the liver, all of the *S. schlegelii* cathepsins were significantly up-regulated at most time points. In detail, CTSBb, CTSC, CTSHa, CTSK, CTSLb, CTSSa, and CTSSb were significantly up-regulated at all time points. CTSSb was up-regulated the most, with an expression level change of 87.43-fold at 48 h, while CTSZb was down-regulated the most, with an expression level change of 5.04-fold at 72 h. Notably, the expression patterns of *S. schlegelii* cathepsins were quite different in the spleen compared with the gill, head kidney, and liver. In the spleen, CTSHa was significantly up-regulated at all points by more than twenty-fold, whereas CTSZb was significantly down-regulated at all four time points, with a maximum down-regulation fold change of 67.22. Apart from this, CTSHa was up-regulated in the four tissues at all time points.

### 2.7. PPI Network Construction of Cathepsins

A PPI network analysis was conducted to reveal potential interacting proteins and immune-related signal transduction pathways. On the one hand, cathepsins were linked within families (Figure 13). For instance, as shown in Figure 13A, CTSAa was predicted to interact with CTSBa, CTSD, and CTSZ, indicating potential connectivity within *S. schlegelii* cathepsin families. Similar PPI patterns were also observed for other members, such as CTSAb, CTSBa, CTSBb, CTSC, CTSDa, CTSHa, CTSZa, and CTSZb. In addition, a few immune-related genes were also observed to show high connectivity to *S. schlegelii* cathepsins, including interleukin genes (IL12RB2), TLR (TLR3), TOLL (toll8), CD genes (CD74 and CD74a), mmp (MMP13b and MMP20a,) mhc (MHC2A, MHC2B, MHC2BL, MHC2DAB, and MHC2DCB), bcl (BCL2, BCL2b, and BCL2L1), and irf (IRF6) (Figure 13).

## 3. Discussion

In the present study, we systematically identified a complete repertoire of eighteen cathepsin genes in the *S. schlegelii* genome and transcriptome datasets. Firstly, multiple alignments of the cathepsin superfamily were performed based on the obtained sequences, revealing that most of the common features were present, including the high-level conservation of the glutamine oxyanion hole (except for cathepsins Aa, Ab, Da, and Db) and cysteine, histidine (except for cathepsins Aa, Ab, Da, Db, and Hb), and asparagine active site residues (except for cathepsins Aa, Da, Db, and Hb), which play important roles in the formation and stabilization of the catalytic sites of activating enzymes. These results are identical to the characteristics of the cathepsin family [38]. Secondly, based on the *S. schlegelii* genome and transcriptome datasets, we performed gene structure, phylogenetic, syntenic, and evolutionary rate analyses and determined their expression patterns after infection with *A. salmonicida*. These results are expected to be valuable for comparative immunological studies and suggest that members of the cathepsin family in *S. schlegelii* play vital roles in immune responses to bacterial infection in teleost species.

By comparing the number of cathepsins identified in *S. schlegelii* and other teleost species, combined with the evolutionary rate analysis, several interesting phenomena in the evolution process were observed. On the one hand, a few members (cathepsins A, D, H, and Z) were duplicated through whole-genome duplications. Admittedly, the cathepsin superfamily is comparatively conserved, but there are also unique differences in various taxonomic species, suggesting that the function and evolution mechanism of the cathepsin superfamily need to be further studied in different teleost species. On the other hand, the lowest Ka/Ks ratios were observed in CTSBa (0.13) and CTSBb (0.14), and the highest Ka/Ks ratios were observed in CTSZa and CTSZb, with values of 1.97 and 1.75, respectively. The Ka/Ks ratios reveal that they underwent efficient purifying selection.

The immune system of fish is very different from that of mammals. Fish lack bone marrow and lymph nodes and instead use the kidney as the main lymphoid organ [39]. In addition, the major lymphoid tissues in teleost fishes are the kidney, thymus, spleen, and mucosa-associated lymphoid tissues, which contain lymphocytes, macrophages, and various types of granulocytes [40]. In this study, we explored the expression profiles of eighteen cathepsin genes in *S. schlegelii*. On the one hand, all cathepsins were expressed constitutively in various tested tissues, suggesting that they may play multifunctional roles in healthy *S. schlegelii*. Among them, the highest expression level was observed in the spleen, followed by the gill, intestine, head kidney, and muscle. The kidney and spleen are important hematopoietic and lymphoid organs that are closely related to the immune response in fish [41]. Notably, cathepsins F, La, and Hb were highly expressed in muscle (Figure 8). A previous study reported that using antibodies directed against chum salmon cathepsins B and L showed that these enzymes were derived from muscle-invading phagocytes [42]. Hence, the differential expression of cathepsins F, La, and Hb in *S. schlegelii* muscle may be related to muscle degradation in fish. On the other hand, *Ss*CTSK was widely expressed in all of the examined tissues, with higher expression levels observed in the spleen, head kidney, gill, and skin, consistent with previous studies in turbot [23,43,44]. There are two copies of CTSH in *S. schlegelii*. CTSHa was expressed at a high level in the intestine, indicating that CTSHa might be involved in intestinal immune responses [45]. In contrast, CTSHb was highly expressed in muscle. Interestingly, CTSH was highly expressed in the intestine and differentially expressed in muscle at the same time in rock bream [46]. A high expression level of CTSBb was observed in the intestine, which is consistent with the highest expression level of CTSB in the intestine of miiuy croaker (*Miichthys miiuy*) [38], whereas CTSBa showed a high expression level in the spleen, consistent with the high expression level of CTSB in the spleen of Nile tilapia (*Oreochromis niloticus*) [47]. The difference in cathepsin expression in different aquatic animals may be related to the different species, living environments, and physiological statuses of individuals [48].

In order to explore the immune roles of cathepsins, the expression profiles of cathepsins following bacterial infection were characterized in the head kidney, spleen, liver, and gill. After *A. salmonicida* infection, most cathepsins were significantly differentially expressed, and obvious tissue-specific and time-dependent expression patterns were observed, suggesting their involvement in response to bacterial infection. In general, the expression patterns of cathepsins were similar among the gill, head kidney, and liver, exhibiting a general up-regulation trend at most time points after *A. salmonicida* infection (Figure 9, Figure 10 and Figure 11). However, a few cathepsin members, such as CTSBb, CTSF, CTSHb, CTSO, CTSSb, and CTSZb, were observed to be down-regulated at 48 h and 72 h in the spleen (Figure 12). Notably, CTSHa was expressed at high levels in the intestine, spleen, head kidney, and gills of healthy rockfish, and it was also observed to be expressed at high levels in the spleen, head kidney, and gills following *A. salmonicida* infection, indicating the potentially vital role of CTSHa in response to bacterial infection. It was reported that CTSA was significantly up-regulated in gill following *V. anguillarum* infection in turbot [49], which is consistent with our result that CTSAa and CTSAb were up-regulated the most in the gill compared with other tissues after bacterial infection. Similar to what was reported in flounder and turbot, CTSB showed the strongest up-regulation, and the up-regulation of CTSBa in mucosal surfaces could be a result of the inflammatory response against invading pathogens [50,51].

Cysteine proteases, which are the largest group, are synthesized as inactive precursors, and the typical domain architectures include a signal peptide, a propeptide, and a mature peptide. The characteristic ERFNIN motif of endopeptidase cathepsins was found in the propeptide regions of CTSBa, CTSBb, CTSC, CTSF, CTSHa, CTSHb, CTSK, CTSLa, CTSLb, CTSO, CTSSa, CTSSb, CTSZa, and CTSZb. A previous study reported that the ERFNIN signature sequence in CTSH might participate in the scaffold of a helical structure in the pro-region, which was significant for its specific inhibition [52]. The mature form of CTSZ has an ECD motif (Glu-Cys-Asp), which might play an important role in integrin-mediated signal transduction [53]. In addition, CTSAa and CTSAb of *S. schlegelii* are representative members of serine proteases. Previous research has demonstrated that CTSA has both catalytic and protective functions [54]. Besides its enzymic functions, cathepsin A plays an important role in the protection of lysosomal β-galactosidase and neuraminidase against intralysosomal proteolysis by forming a macromolecular complex [55,56]. Moreover, CTSDa and CTSDb of *S. schlegelii* are representative members of aspartic proteases. Previous studies reported that CTSD prefers an α-helical folding of its substrate for proteolytic degradation, and the activity of CTSD may be crucial in determining which peptide products arise from antigen processing and then serve as T-cell epitopes. This information is relevant to the design of synthetic immunogens and vaccines that aim at T-cell activation [57]. In general, cathepsins play physiological roles in protein degradation/turnover (CTSB, CTSL, and CTSH), bone resorption (CTSK), proenzyme activation (CTSB and CTSC), antigen presentation/processing (CTSF, CTSH, CTSL, CTSS, and CTSV), epidermal homeostasis (CTSL), and hormone maturation (cathepsins B and L) [58].

PPI analyses were conducted to further reveal the regulatory mechanisms of *S. schlegelii* cathepsins, which are essential for predicting activities for most proteins. As shown in Figure 13, a number of immune-related genes were observed to show high connectivity to cathepsins, including interleukin genes (IL12RB2), TLR (TLR3), TOLL (toll8), CD genes (CD74 and CD74a), mmp (MMP13b and MMP20a), mhc (MHC2A, MHC2B, MHC2BL, MHC2DAB, and MHC2DCB), bcl (BCL2, BCL2b, and BCL2L1), irf (IRF6) (EIF2B1), chemokines (ccr12.2), etc. CCR12a and CCR12b were both differentially expressed after *A. salmonicida* infection in *S. schlegelii*, suggesting their key roles in immune regulation in fish [59,60]. TLR3-mediated signaling is important for host defense against RNA viruses [61]. Infection with the IHN virus and the time in culture both appeared to enhance MHC class II expression in rainbow trout [62]. In mammals, cell-mediated cytotoxicity is largely mediated by MHC class I molecules, but peptide-loaded MHC class II molecules can also stimulate cytotoxicity by T cells [63,64]. More experimental evidences are required to discover their immune response mechanisms.

## 4. Materials and Methods

### 4.1. Gene Identification

In order to identify cathepsins in turbot, all available sequences of cathepsins from zebrafish, channel catfish, flounder, medaka, tilapia (*Oreochromis niloticus*), fugu, frog (*Xenopus laevis*), chicken (*Gallus gallus*), mouse (*Mus musculus*), and human (*Homo sapiens*) were retrieved from the NCBI (http://www.ncbi.nlm.nih.gov/, accessed on 17 September 2021), Ensemble (http://www.ensembl.org, accessed on 17 September 2021), UniProt (http://www.uniprot.org/, accessed on 17 September 2021), and ZFIN (http://zfin.org/, accessed on 17 September 2021) datasets. These sequences were used as queries to search against the *S. schlegelii* genome [65] and transcriptome datasets [66] using the TBLASTN program with an E-value cutoff of 1 × 10^−5^. Then, Clustal Omega (http:// www.ebi.ac.uk/Tools/msa/clustalo/, accessed on 25 September 2021) was used to eliminate duplicates in the initial sequence pool, and each sequence was confirmed to be a unique gene according to its location on the genome. After that, the ORF (opening reading frame) finder (http://www.ncbi.nlm.nih.govgorf/gorf.html, accessed on 27 September 2021) was used to predict coding sequences, which were further validated by BLASTP against NCBI non-redundant (nr) protein database. Lastly, the Fgenesh program of Molquest software (Softberry Int, version 2.4, Mount Kisco, NY, USA) was used to predict the genes from the retrieved genomic chromosome sequences [67]. Combining the Fgenesh predictions with the blast results captured the complete set of cathepsins’ sequences.

### 4.2. Sequence Analysis

To explore the characteristics of cathepsins, a series of bioinformatic analyses were conducted. Firstly, the lengths of mRNA and amino acid sequences, molecular weights (MWs), and isoelectric points (PIs) of cathepsins were calculated using ProtParam (https://web.expasy.org/protparam/, accessed on 8 October 2021) [68]. ProtComp 9.0 (http://www.softberry.com, accessed on 12 October 2021) was used to predict the subcellular localizations of cathepsins. In addition, the functional domains (motif patterns) were predicted using the simple modular architecture research tool (SMART; http://smart.emblheidelberg.de/, accessed on 15 October 2021) and further endorsed by conserved domains predicted through BLASTP. The number of exons was extracted from the annotation results of the *S. schlegelii* genome reference [66], and the Gene Structure Display Server 2.0 (GSDS, http://gsds.gao-lab.org/, accessed on 26 October 2021) was used to draw the gene structures of cathepsins. The mappings of cathepsins on chromosomes were analyzed by TBtools. The presumed 3D protein structural model was established using PHYRE2 Protein Fold Recognition Server (Phyre2 server) (http://www.sbg.bio.ic.ac.uk/phyre2/html/page.cgi?id=index, accessed on 5 November 2021) [69]. To determine the PPI network of expressed proteins, amino acid sequences of cathepsins were blasted against *C. semilaevis* or *D. rerio* by using STRING software 11.0. Representation of the protein–protein network was analyzed at a confidence score of 0.40 from text mining, experiments, databases, co-expression, and neighborhood sources [70].

### 4.3. Phylogenetic Analysis

Phylogenetic analysis was conducted using the amino acid sequences of cathepsins, along with several representative vertebrates, including human, mouse, chicken, fugu, zebrafish, channel catfish, flounder, rainbow trout (*Onchorynchus mykiss*), tongue sole (*Cynoglossus semilaevis*), Atlantic salmon (*Salmo salar*), and turbot. Alignment of multiple amino acid sequences with default parameters was conducted by MUSCLE (multiple sequence comparison by log-expectation) [71]. Then, the phylogenetic trees of cathepsins were constructed in the MEGA 6 program. In detail, the phylogenetic tree of cathepsin was reconstructed with JTT (Jones–Taylor–Thornton) + G (gamma distribution for modeling rate heterogeneity) + I (invariant sites) + F (Freqs) model, with a proportion of invariable sites of 0.01 and Gamma shape parameter of 2.27. Bootstrapping with 1000 replications was performed to test the phylogenetic tree, and gaps/missing data treatment was analyzed using all sites. Finally, EvolView (http://www.evolgenius.info/evolview, accessed on 15 November 2021) (http://www.evolgenius.info/evolview, accessed on 18 November 2021) was used to annotate the phylogenetic trees.

### 4.4. Synteny and Schematic Genomic Organization Analysis

Syntenic analysis was conducted to provide additional evidence for the identification of cathepsins based on the comparison of neighboring genes of cathepsins with those of zebrafish and channel catfish (or turbot). The flanking genes of *S. schlegelii* cathepsins were predicted from *S. schlegelii* genomic chromosomes using the Fgenesh program in Molquest software [67], and the identified amino acid sequences were annotated by running BLASTP against NCBI non-redundant (nr) database and UniProt Knowledgebase (UniProtKB). Meanwhile, the conserved syntenic blocks of cathepsins and their flanking genes in zebrafish and channel catfish (or turbot) were obtained from the NCBI database. Zebrafish is a model organism and has the most updated nomenclature system (http://zfin.org/, accessed on 25 November 2021), so we named cathepsins after zebrafish whenever possible based on orthologs, which were determined by phylogenetic and syntenic analysis.

### 4.5. Evolutionary Rate Analyses

There are two indicators of base substitution frequency: synonymous sites (Ks) and non-synonymous substitution sites (Ka). Their ratios (Ka/Ks) are used to measure the evolutionary rates of genes. When the ratio is >1, it indicates that these genes underwent positive selection, while when the ratio is <1, it indicates that these genes underwent purifying selection (<1) during the evolutionary process. To illustrate the evolutionary rates of cathepsins among the teleost species, the blast method was used to extract the homologs or regions containing cathepsins in available species, including zebrafish, channel catfish, turbot, fugu, medaka, and tilapia. Subsequently, the nucleic acid sequences of cathepsins identified from the selected teleost species were imported into the MEGA software (version: 6.0, Center for Evolutionary Medicine and Informatics, Tampe,FL, USA) and then aligned according to codons with the parameters set as: gap opening penalty: 10; gap extension penalty: 0.2; delay divergent cutoff: 30%. Finally, DnaSP ver. 5 was used to calculate the ratios of Ka/Ks [72].

### 4.6. Sample Collection of Healthy S. schlegelii

In order to characterize the basal expression patterns of cathepsins in *S. schlegelii*, nine tissues of healthy *S. schlegelii* were collected. Experimental *S. schlegelii* adults were obtained from a local fish farm in Qingdao, Shandong Province, with an average length of 15 ± 2 cm. Before sample collection, fish were reared in the laboratory and acclimatized in a flow-through system filtered with seawater for one week without feeding. The DO, pH, and temperature were 5.2 ± 0.35 mg/L, 8.2 ± 0.01, and 22.03 ± 0.58 °C, respectively. Thirty fish were anesthetized with tricaine methanesulfonate (MS-222, 100 mg/L) in seawater. Then, nine tissues, namely, the blood, brain, head kidney, skin, gill, muscle, spleen, intestine, and liver, were collected, flash-frozen in liquid nitrogen, and then stored at −80 °C in an ultra-low freezer until the preparation of RNA.

### 4.7. Bacterial Challenge and Sample Collection

In order to characterize the immune response of cathepsins to bacterial infection, *A. salmonicida* was selected to conduct the injection challenge. The experimental fish were the same as mentioned above. Thereafter, a pre-challenge in *S. schlegelii* was performed. Subsequently, confirmed *A. salmonicida* isolated from symptomatic fish was cultured in LB medium at 28 °C overnight at 180 rpm/min. The aquaria, with 30 fish in each aquarium, were randomly assigned to 6 h, 24 h, 48 h, and 72 h post-injection groups. Then, the experimental fish were intraperitoneally injected with 0.1 mL *A. salmonicida* at a concentration of 1 × 10^7^ CFU/mL. At the same time, the fish in the control group were injected with an equal amount of physiological saline. During the challenge, the fish were collected after being anesthetized with MS-222 (100 mg/L). Subsequently, the tissues (head kidney, liver, spleen, and gill) were collected after injection at different time points (6 h, 24 h, 48 h, and 72 h) and were designated as AS_6 h, AS_24 h, AS_48 h, and AS_72 h, respectively. Each group had three replicates, and each replicate included five random individuals. All samples were flash-frozen in liquid nitrogen and then stored in an ultra-low freezer at −80 °C until RNA extraction.

### 4.8. Total RNA Extraction and Real-Time PCR Analysis

Before RNA extraction, a mortar and pestle were used to homogenize samples under liquid nitrogen. Using Trizol Reagent (Invitrogen, Carlsbad, CA, USA), the total RNA was extracted according to the supplied instructions. Then, the integrities of RNAs were detected on 1% agarose gel. Then, all RNA samples were run on the gel to ensure that no genomic DNA existed. Using Nanodrop 2000 (Thermo electronic North America LLC, Boston, FL, USA), the RNA quality and quantity of each sample were determined. All extracted samples had an A260/280 ratio greater than 1.8. A NanoPhotometer spectrophotometer (IMPLEN, Westlake Village, CA, USA) was used to check RNA concentration.

Quantitative real-time PCR (qPCR) was conducted to examine the expression patterns of cathepsins in different tissues (head kidney, liver, spleen, and gill) following *A. salmonicida* challenges. Gene-specific primers were designed using PrimerQuest (https://sg.idtdna.com/PrimerQuest/Home, accessed on 20 December 2021) based on the gene sequences of cathepsin (Table 2). The β-actin gene was used as an internal control for normalization of the expression levels (Table 2). First-strand cDNA (500 ng RNA per 10 μL reaction) was synthesized according to the manufacturer’s protocol, and the PrimeScript™ RT reagent Kit (Takara, Otsu, Japan) was used to synthesize first-strand cDNA (500 ng RNA per 10 μL reaction). After that, qPCR was performed on a CFX96 real-time PCR detection system (Bio-Rad Laboratories, Hercules, CA, USA) following the manufacturer’s instructions. The reactions were performed in 20 μL volumes containing 2 μL of the diluted template cDNA, 0.6 μL of each primer, 10 μL of SYBR^®^ Premix Ex TaqTM II (TliRNaseH Plus), and 6.8 μL of RNA-free water. The thermal cycling profile was performed as follows. In order to verify the specificity of the amplicons, the PCR reaction mixture was denatured at 95 °C for 30 s, followed by 40 cycles of 95 °C for 30 s, 58 °C for 30 s, followed by dissociation curve analysis, 5 s at 65 °C, and then up to 95 °C at a rate of 0.1°C/s increment. A non-template control was performed as a negative control for each gene, and each reaction was confirmed by repeating the qPCR analysis in triplicate. Finally, the 2^−ΔΔCt^ method was used to calculate the relative gene expression fold changes [73].

## 5. Conclusions

In summary, genome-wide identification of eighteen cathepsins was performed in *S. schlegelii*. Gene structure, phylogenetic, synteny, genome organization, and evolution rate analyses allowed annotations of these genes, and the results provide insights into their evolutionary dynamics. Furthermore, cathepsins were ubiquitously expressed in nine examined tissues, with high expression levels observed in the head kidney, spleen, liver, and gill. Meanwhile, most cathepsins were differentially expressed in four examined tissues after *A. salmonicida* infection, suggesting their participation both in homeostasis and in immune responses to bacterial infection. Finally, PPI analyses indicate that cathepsins interact with a few immune-related genes, such as interleukin, chemokine, MHCII, and TLR genes. These results indicated the vital roles of cathepsins in teleost immunity, and further studies are needed to explore the immune function and mechanism of cathepsins in teleost host immune activities.

## Figures and Tables

**Figure 1 marinedrugs-20-00504-f001:**
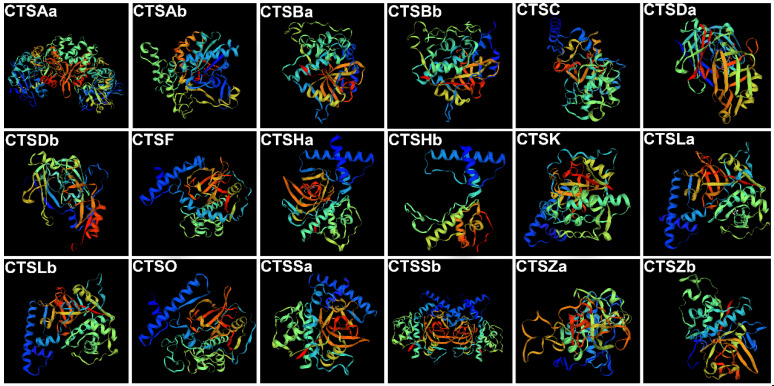
The 3D structural models of *S. schlegelii* CTSAa, CTSAb, CTSBa, CTSBb, CTSC, CTSDa, CTSDb, CTSF, CTSHa, CTSHb, CTSK, CTSLa, CTSLb, CTSO, CTSSa, CTSSb, CTSZa, and CTSZb were predicted using Phyre2. The images were colored by rainbow from the N- to C-terminus, and 95% of residues were modeled at >90% confidence.

**Figure 2 marinedrugs-20-00504-f002:**
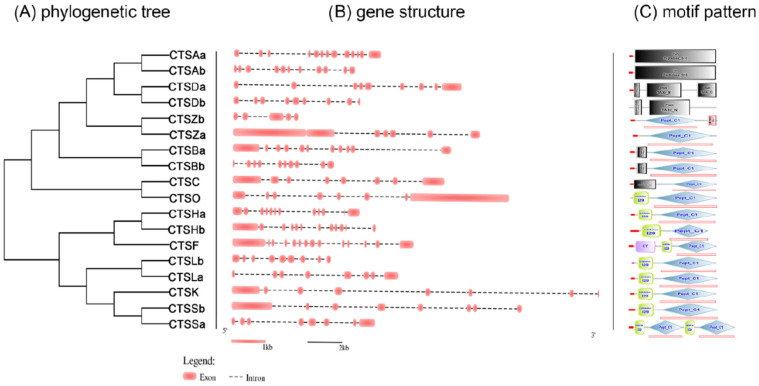
Gene structure of *S. schlegelii* cathepsin genes. (**A**) indicates the phylogenetic tree of black rockfish cathepsin genes (**B**) indicates the gene structures of each gene. (**C**) indicates the motif pattern of each gene. The pink ellipses indicate the exons of cathepsin genes. The dotted lines indicate introns. The green rectangles and blue rectangles indicate the inhibitor 129 and Pept_C1 domains, respectively.

**Figure 3 marinedrugs-20-00504-f003:**
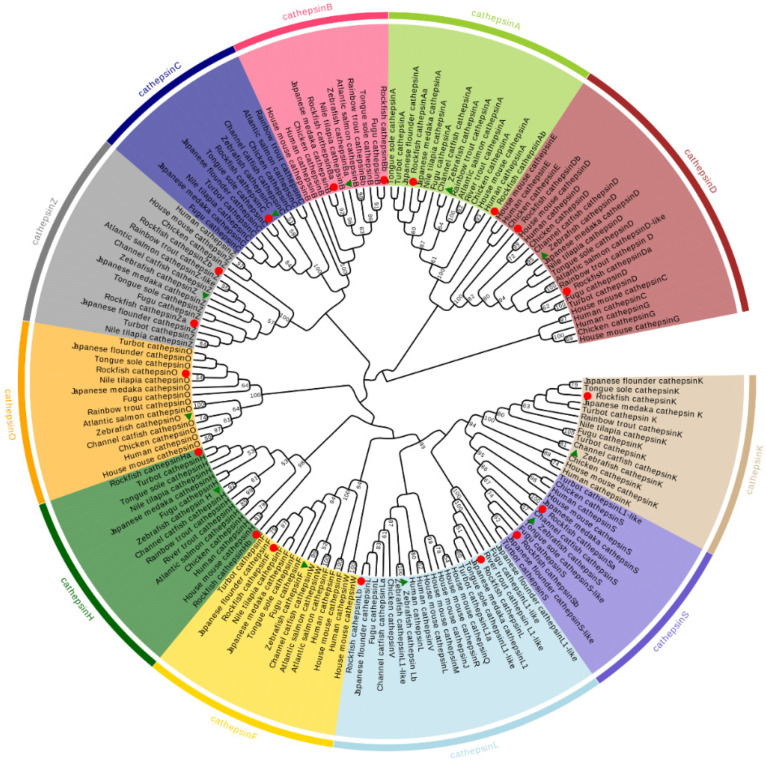
Phylogenetic analysis of *S. schlegelii* cathepsin genes. The phylogenetic tree was constructed based on the amino acid sequences of cathepsin of vertebrates using the neighbor-joining method in MEGA 6. Gaps/missing data treatment was analyzed using all sites, and the phylogenetic tree was evaluated with 1000 bootstrap replications. The bootstrap values are indicated by numbers at the nodes. The red circles indicate the *S. schlegelii* cathepsin genes characterized in the present study, and green triangles indicate zebrafish cathepsin genes.

**Figure 4 marinedrugs-20-00504-f004:**
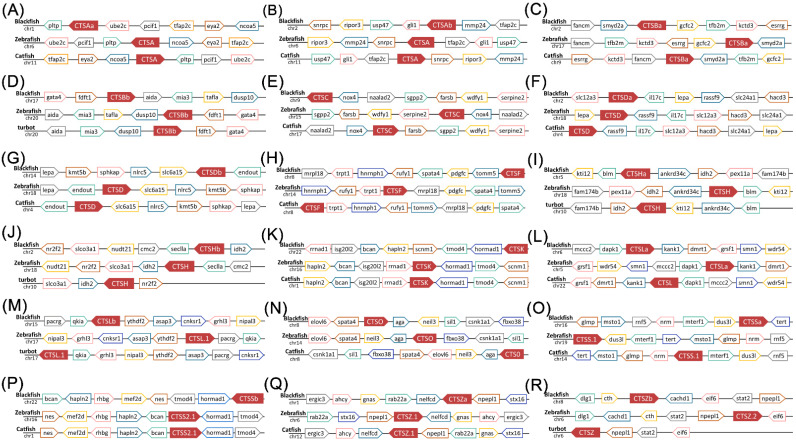
Syntenic analysis of *S. schlegelii* cathepsin genes with channel catfish and zebrafish. (**A**) CTSAa; (**B**) CTSBb; (**C**) CTSBa; (**D**) CTSBb; (**E**) CTSC; (**F**) CTSDa; (**G**) CTSDb; (**H**) CTSF; (**I**) CTSHa; (**J**) CTSHb; (**K**) CTSK; (**L**) CTSLa; (**M**) CTSLb; (**N**) CTSO; (**O**) CTSSa; (**P**) CTSSb; (**Q**) CTSZa; and (**R**) CTSZb. These syntenies were generated with information obtained from the NCBI. Full gene names are provided in Appendix A.

**Figure 5 marinedrugs-20-00504-f005:**
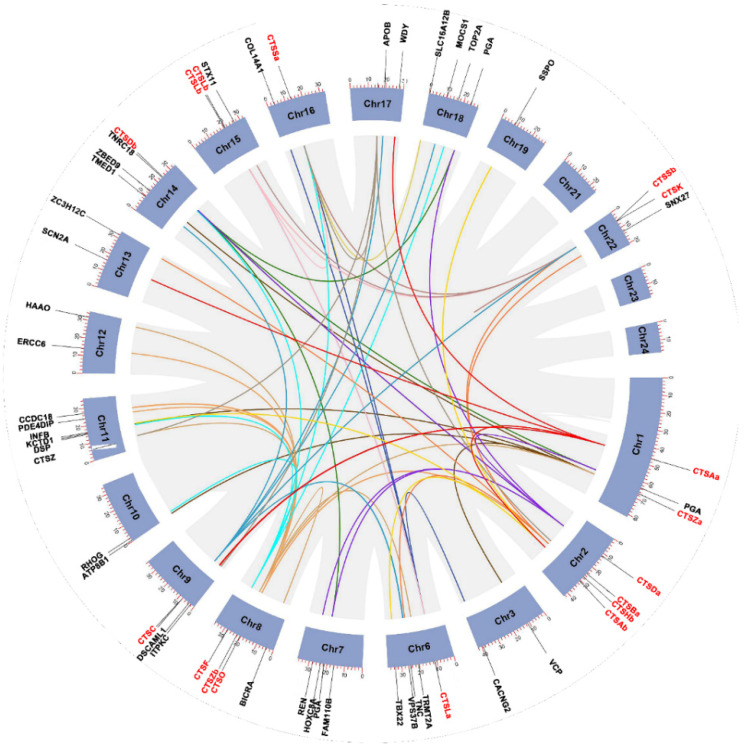
Chromosomal distribution and interchromosomal relationship of *S. schlegelii* cathepsin genes. Gray lines indicate all synteny blocks in the *S. schlegelii* genome, and different colored lines indicate duplicated cathepsin family gene pairs. The chromosome number is labeled on the top of each chromosome. Blue rectangles represent *S. schlegelii* chromosomes 01–24. The identified cathepsin genes are listed on the outermost edge of thermal maps.

**Figure 6 marinedrugs-20-00504-f006:**
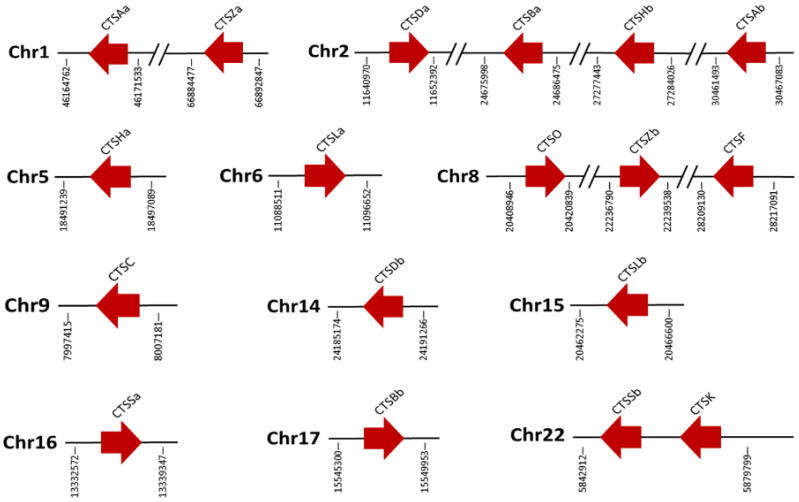
Schematic genomic organization of *S. schlegelii* cathepsin genes. Red arrowheads indicate their transcriptional orientation. The numbers under the cathepsin genes indicate their position on the genome.

**Figure 7 marinedrugs-20-00504-f007:**
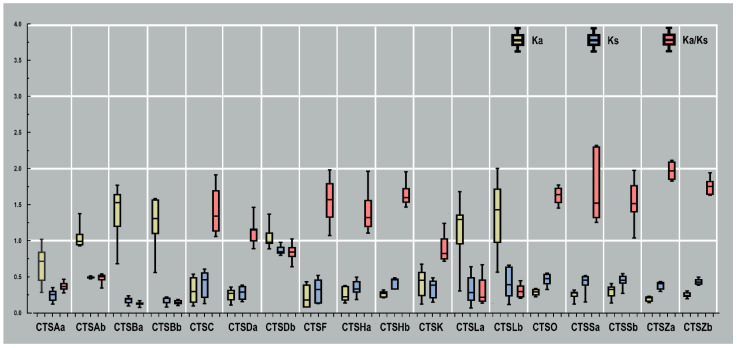
The evolutionary rate comparison of *S. schlegelii* cathepsin genes in selected teleost species. Ka values represent the non-synonymous rates, which are marked in yellow, and Ks values represent the synonymous rates, marked in blue. The ratios of Ka/Ks are marked in pink. The box plot represents the standard deviation (SD) and median and shows values between the first quartile and the third quartile of Ka, Ks, and Ka/Ks of each gene.

**Figure 8 marinedrugs-20-00504-f008:**
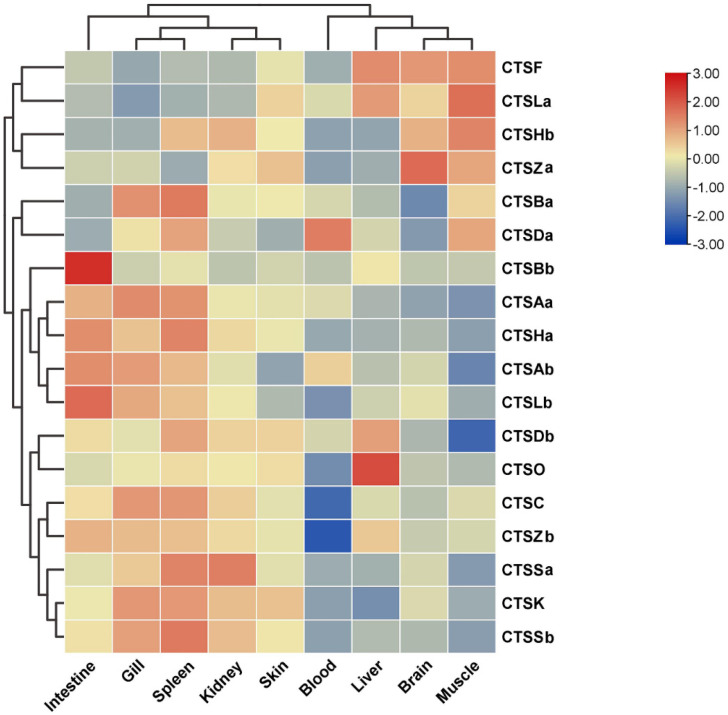
Tissue expression analysis of cathepsin genes in healthy *S. schlegelii*. Relative expression values were calculated using blood as the reference (set as 1) normalized to the expression of the β-actin rRNA reference gene.

**Figure 9 marinedrugs-20-00504-f009:**
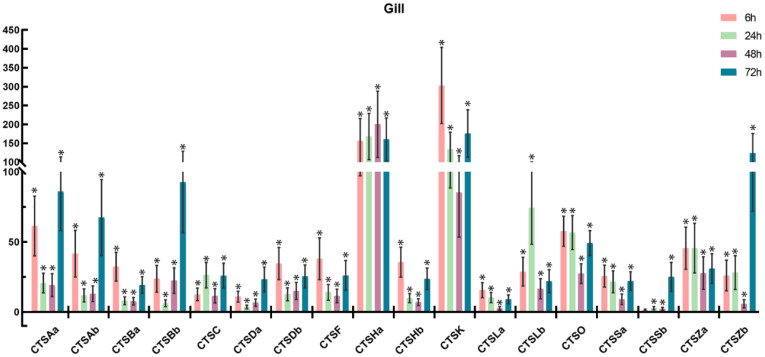
Expression of *S. schlegelii* cathepsin genes in the gill after *A. salmonicida* infection at 6 h, 24 h, 48 h, and 72 h. The *y*-axis indicates fold change, and the *x*-axis provides the names of the genes. The significantly differentially expressed genes (|fold change| ≥ 2, with the *p*-value < 0.05) were marked with *.

**Figure 10 marinedrugs-20-00504-f010:**
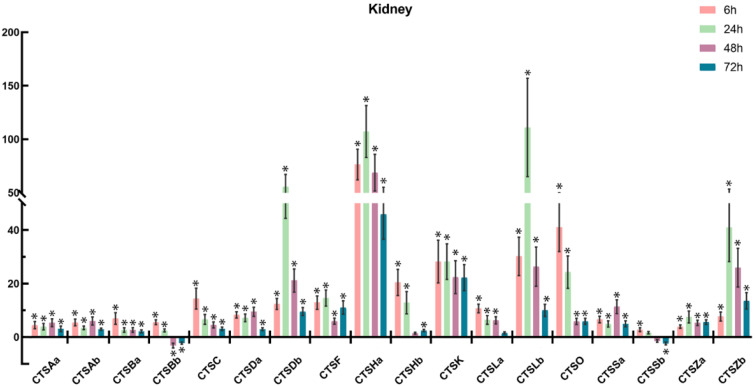
Expression of *S. schlegelii* cathepsin genes in the head kidney after *A. salmonicida* infection at 6 h, 24 h, 48 h, and 72 h. The *y*-axis indicates fold change, and the *x*-zxis provides the names of the genes. The significantly differentially expressed genes (|fold change| ≥ 2, with the *p*-value < 0.05) were marked with *.

**Figure 11 marinedrugs-20-00504-f011:**
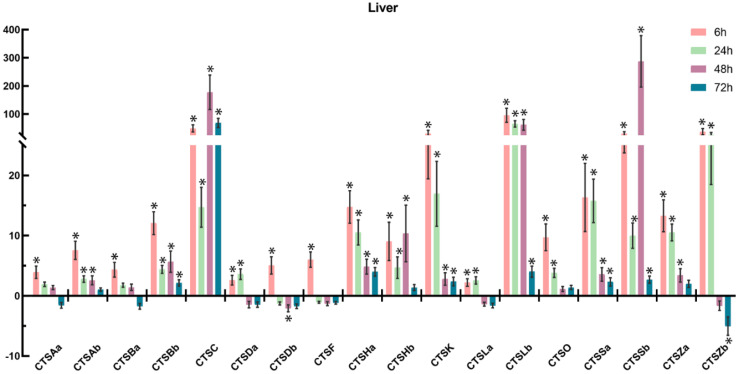
Expression of *S. schlegelii* cathepsin genes in the liver after *A. salmonicida* infection at 6 h, 24 h, 48 h, and 72 h. The *y*-axis indicates fold change, and the *x*-axis provides the names of the genes. The significantly differentially expressed genes (|fold change| ≥ 2, with the *p*-value < 0.05) were marked with *.

**Figure 12 marinedrugs-20-00504-f012:**
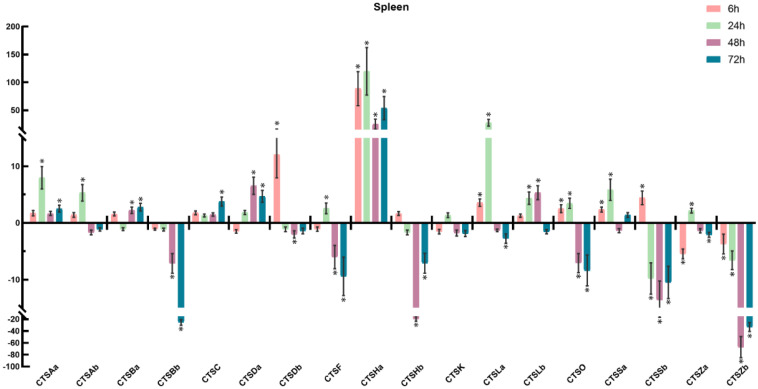
Expression of *S. schlegelii* cathepsin genes in the spleen after *A. salmonicida* infection at 6 h, 24 h, 48 h, and 72 h. The *y*-axis indicates fold change, and the *x*-axis provides the names of the genes. The significantly differentially expressed genes (|fold change| ≥ 2, with the *p*-value < 0.05) were marked with *.

**Figure 13 marinedrugs-20-00504-f013:**
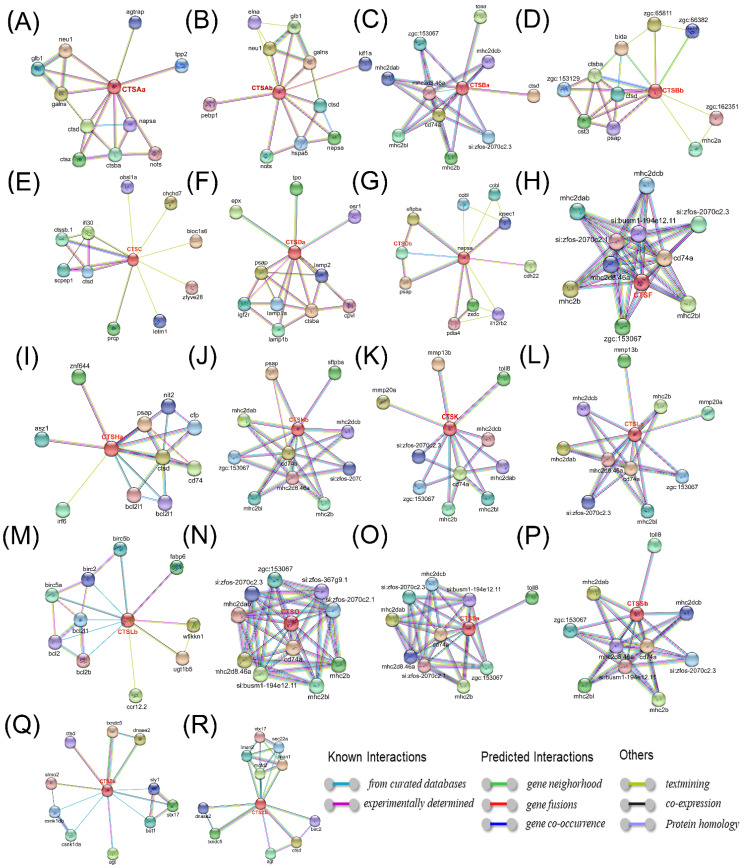
Protein–protein interactions that show predicted functional partners of *S. schlegelii* (**A**) CTSAa; (**B**) CTSBb; (**C**) CTSBa; (**D**) CTSBb; (**E**)CTSC; (**F**) CTSDa; (**G**) CTSDb; (**H**) CTSF; (**I**) CTSHa; (**J**) CTSHb; (**K**) CTSK; (**L**) CTSLa; (**M**) CTSLb; (**N**) CTSO; (**O**) CTSSa; (**P**) CTSSb; (**Q**) CTSZa; and (**R**) CTSZb. Predicted by STRING 11.0 with the setting interaction evidence as network edges.

**Table 1 marinedrugs-20-00504-t001:** Identification of cathepsin genes in *S. schlegeli*.

Genes	CDS (aa)	PI	MW (kDa)	Ori	Chr	Location	Instability Index	Aliphatic Index	Formula
CTSAa	472	5.26	53.32	−	1	46,164,762–46,171,533	34.42	84.62	C_2408_H_3630_N_624_O_706_S_22_
CTSAb	459	5.41	51.37	−	2	30,461,493–30,467,083	29.89	74.34	C_2333_H_3442_N_600_O_679_S_19_
CTSBa	330	5.38	36.06	−	2	24,675,998–24,686,475	32.27	69.42	C_1599_H_2412_N_424_O_486_S_22_
CTSBb	330	5.20	36.06	+	17	15,545,300–15,549,953	31.46	70.94	C_1590_H_2411_N_433_O_487_S_21_
CTSC	454	5.78	50.63	−	9	7,997,415–8,007,181	33.78	70.13	C_2282_H_3423_N_591_O_656_S_31_
CTSDa	396	6.88	42.50	+	2	11,640,970–11,652,392	35.19	89.65	C_1936_H_3023_N_501_O_573_S_16_
CTSDb	339	8.41	36.97	−	14	24,185,174–24,191,266	40.74	94.07	C_1665_H_2617_N_437_O_483_S_15_
CTSF	474	5.57	53.02	−	8	28,209,130–28,217,091	39.98	76.96	C_2354_H_3652_N_628_O_715_S_26_
CTSHa	325	5.53	36.41	−	5	18,491,239–18,497,089	35.70	60.00	C_1614_H_2404_N_430_O_494_S_21_
CTSHb	215	5.47	24.66	−	2	27,277,443–27,284,026	42.11	67.53	C_1110_H_1650_N_290_O_326_S_12_
CTSK	330	5.56	36.41	−	22	5,860,447–5,879,799	32.01	72.39	C_1592_H_2468_N_444_O_490_S_23_
CTSLb	333	5.02	36.63	−	15	20,462,275–20,466,600	34.81	74.11	C_1610_H_2471_N_435_O_508_S_18_
CTSLa	336	5.62	38.08	+	6	11,088,511–11,096,652	29.14	66.43	C_1688_H_2558_N_462_O_511_S_18_
CTSO	335	8.06	37.27	+	8	20,408,946–20,420,839	32.89	75.07	C_1664_H_2531_N_461_O_489_S_14_
CTSSa	640	8.00	71.66	+	16	13,332,572–13,339,347	42.08	80.86	C_3192_H_4925_N_891_O_924_S_34_
CTSSb	337	5.72	37.17	−	22	5,842,912–5,857,384	35.35	66.32	C_1629_H_2496_N_448_O_503_S_24_
CTSZa	357	6.72	39.68	−	1	66,884,477–66,892,847	41.59	67.45	C_1746_H_2660_N_492_O_533_S_19_
CTSZb	301	6.44	33.46	+	8	22,236,790–22,239,538	31.76	67.01	C_1497_H_2240_N_398_O_440_S_19_

CDS indicates the length of the coding sequence. PI indicates the isoelectric point. MW indicates the molecular weight. Ori + and − indicate the plus and minus strands, respectively. Chr indicates the chromosomal location.

**Table 2 marinedrugs-20-00504-t002:** Primers of *S. schlegelii* cathepsin genes used in this work.

Gene Name	Forward Primer	Reverse Primer
β-actin	GTGCGTGACATCAAGGAGAAGC	TGTTGTAGGTGGTCTCGTGGA
CTSAa	GAGCTGGTGATGAGAGATCTTG	GACGGACGTTCAGGTTTAGAG
CTSAb	GTGAATGTGGCCTTTGGTATTG	GGGTGTTTCCTGTCGTTCTTA
CTSBa	CACTCCCAGCTACAAAGTAGAC	CTACTGGGCCGTTCTTGTATAG
CTSBb	TGGGCTGTTATGGTGGTTATC	AGCCGACATTGGAGTTATACAG
CTSC	GTACACTGAAGATGGCCCTAAG	TGGAGGTACTGGTTTCACTTTC
CTSDa	GCTTGCTTCACCGCAAATATAA	GACAGACTGCCACTTCCATAC
CTSDb	CCCTCCATTCACTGCTCTTT	GGAGAACTCTGTGCCATTCTT
CTSF	GGGATCCGCTACACCATAAC	GTTTCTGCAACTCGGGATAGA
CTSHa	CCTCTGACTTCATGCACTACTC	TGCCGTTCTCTTGTCCATAC
CTSHb	GCTCTTCCTACCCTCTACCTTTA	CGTGTGTGTGTATGTTGTCTCT
CTSK	GGAGGAGGATACATGACCAATG	GCCATGCCTGTTGAATTGTAG
CTSLa	ACGGAGGGATAGACACAGAA	CGTCACCTTGTTTCACATCAAC
CTSLb	GGCTGGCACAGTAGGAAATA	CATTGAATGCTCCAGGTTGTG
CTSO	CCTCAGTCAGAGTATCCCTACA	CACTGAAGTCATGTGCAGTAAAG
CTSSa	GTGGAAGAAGACACACGAGAAG	AGGTTGTGCATGGTGATGAG
CTSSb	CTGGGCACTATGGAAGAAGATG	CCAAGGAGGTTTCCAGGTTATG
CTSZa	GGCAGATCGCATCAACATTAAG	GATCTCCTCCATGGCAACTAC
CTSZb	GAACCAGCACATCCCTAAGT	CGTGTTGGACGGACAGATAA

## Data Availability

Not applicable.

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
