# Peer review of "Genome-Wide Identification, Evolutionary Analysis, and Expression Patterns of Cathepsin Superfamily in Black Rockfish (Sebastes schlegelii) following Aeromonas salmonicida Infection"

_marinedrugs, 2022, doi:10.3390/md20080504_

Round 1
Reviewer 1 Report
Abstract
This section is well prepared.
Introduction
This section is well prepared.
Material and methods
Which diet was used for maintaining the fish?
Include information of water quality (DO, pH, Temp, Ammonia, etc).
Results
This section is well prepared.
Discussion
This section is well prepared.
I would like the authors to deepen on the possible applications when evaluating cathepsin expression during S. schlegelii culture, particularly when considering different culture conditions and larviculture.
Conclusion
This section is well prepared.
Author Response
Abstract
This section is well prepared.
Response: Thanks very much for your positive comments.
Introduction
This section is well prepared.
Response: Thanks very much for your positive comments.
Material and methods
Which diet was used for maintaining the fish?
Include information of water quality (DO, pH, Temp, Ammonia, etc).
Response: Thanks for your comments. After being purchased from a local fish farm in Qingdao, the experimental fish were reared in the lab and acclimatized for one week before challenge experiment and sampling. The experiment fish were not fed during the acclimation period. Meanwhile, the DO, pH, temperature were 5.2±0.35mg/L, 8.2±0.01, and 22.03±0.58℃, respectively. Accordingly, we revised the methods to elucidate the feeding and rearing information. It now reads “Before sample collection, fish were reared in the laboratory and acclimatized in the laboratory in a flow-through system filtered with seawater for one week without feeding. The DO, pH, and temperature were 5.2±0.35mg/L, 8.2±0.01, and 22.03 ± 0.58℃, respectively.”
Results
This section is well prepared.
Response: Thanks very much for your positive comments.
Discussion
This section is well prepared.
I would like the authors to deepen on the possible applications when evaluating cathepsin expression during S. schlegelii culture, particularly when considering different culture conditions and larviculture.
Response: Thanks for your valuable suggestions for us to improve the discussion part. As you suggested, it would be very meaningful to explore the cathepsin expression at different culture conditions. In this study, S. schlegelii were purchased for fish farming and cultured in our lad for only one week to finish acclimation, and then were used for bacterial challenge and sampling. We did not take the different culture conditions into consideration by now. We will pay attention to the expression signature of cathepsins at different culture condition in our future work.
Conclusion
This section is well prepared.
Response: Thanks very much for your positive comments.

Reviewer 2 Report
The article is interesting to read. However, it needs to address the following comments:
1. The figure quality needs to be improved for readability and clarity i.e Figures 2, 3, 5 etc the label is barely readable. the figure legends need to be comprehensive and understood without the need to go back to the text to understand the figure
2. please define the abbreviations the first time they appear in the text, especially in the abstract
3. please write in vivo and in vitro in italic formatting across the whole manuscript
4. Authors should include a paragraph or a few lines on the reasons for the selection of the cathepsin superfamily in black rockfish?
5. some of the relevant work hasn't been cited for example (https://doi.org/10.1016/j.fsi.2020.05.068 - PMID: 31400512; PMID: 32001355, PMID: 27569982 )
6. in Table 1 please include what ori (-, +) means and GRAVY as well
7. English and grammar need to be checked across the whole manuscript for clarity and typos
8. methods should be detailed and comprehensive please revise accordingly
Author Response
The article is interesting to read. However, it needs to address the following comments:
- The figure quality needs to be improved for readability and clarity i.e Figures 2, 3, 5 etc the label is barely readable. the figure legends need to be comprehensive and understood without the need to go back to the text to understand the figure
Response: Thanks for your comments. We now revised the manuscript by enlarging the figure label of Figure 2, 3, and 5. Meanwhile, we also revised the figure legends and table legends throughout the manuscript in order to be comprehensively understood, according to your suggestion.
- please define the abbreviations the first time they appear in the text, especially in the abstract
Response: Thanks for your comments. We revised the manuscript by defining the abbreviations the first time they appear, such as PPI for protein-protein interaction.
- please write in vivo and in vitro in italic formatting across the whole manuscript
Response: Thanks for your comments. We now revised the manuscript by italicizing the “in vivo” and “in vitro” throughout the manuscript.
- Authors should include a paragraph or a few lines on the reasons for the selection of the cathepsin superfamily in black rockfish?
Response: Thanks for your comments. We revised the manuscript by adding a few lines on the reason for the selection of the cathepsin superfamily in black rockfish. It now reads “As cathepsins exert vital roles in immune responses to bacterial infections, clarification of the molecular mechanism for cathepsins in responses to A. salmonicida infection should promote the development of effective strategies in bacterial disease management for S. schlegelii.”
- some of the relevant work hasn't been cited for example (https://doi.org/10.1016/j.fsi.2020.05.068 - PMID: 31400512; PMID: 32001355, PMID: 27569982 )
Response: Thanks for your comments. The first three references you recommended are relevant to black rockfish cathepsins (https://doi.org/10.1016/j.fsi.2020.05.068 - PMID: 31400512; PMID: 32001355). We revised the manuscript by citing them as 24, 37, and 38 at line 97.
- in Table 1 please include what ori (-, +) means and GRAVY as well
Response: Thanks for your comments. We now revised the figure legends and table legends throughout the manuscript in order to be comprehensively understood, according to your suggestion.
- English and grammar need to be checked across the whole manuscript for clarity and typos
Response: Thanks for your comments. We now revised the whole manuscript to fix typos.
- methods should be detailed and comprehensive please revise accordingly
Response: Thanks for your comments. We now revised the methods carefully, according to your suggestion.
